# Effects of digital devices and online learning on computer vision syndrome in students during the COVID-19 era: an online questionnaire study

Kasem Seresirikachorn,[1] Warakorn Thiamthat,[2] Wararee Sriyuttagrai,[3] Ngamphol Soonthornworasiri,[4] Panisa Singhanetr,[5] Narata Yudtanahiran,[6] Thanaruk Theeramunkong [ID] [1,7]

For numbered affiliations see end of article.

**Correspondence to**
Dr Thanaruk Theeramunkong; thanaruk@siit.tu.ac.th

## ABSTRACT

**Purpose** Computer vision syndrome (CVS) describes a group of eye and vision-related problems that result from prolonged digital device use. This study aims to assess the prevalence and associated factors of CVS among students during the lockdown resulting from the COVID-19 pandemic.

**Methods** A cross-sectional, online, questionnaire-based study performed among high school students in Thailand.

**Results** A total of 2476 students, with mean age of 15.52±1.66 years, were included in this study. The mean number of hours of digital device use per day (10.53±2.99) increased during the COVID-19 pandemic compared with before its advent (6.13±2.8). The mean number of hours of online learning was 7.03±2.06 hours per day during the pandemic. CVS was found in 70.1% of students, and its severity correlated with both the number of hours of online learning and the total number of hours of digital device usage (p<0.001). Multiple logistic regression analysis revealed that the factors associated with CVS included age ≤15 years (adjusted OR (AOR)=2.17), overall digital device usage >6 hours per day (AOR=1.91), online learning >5 hours per day (AOR=4.99), multiple digital device usage (AOR=2.15), refractive errors (AOR=2.89), presence of back pain (AOR=2.06) and presence of neck pain (AOR=2.36).

**Conclusions** The number of hours of digital device usage increased during lockdown. Over 70% of children had CVS, whose associated factors, including hours of digital device usage, hours of online learning, ergonomics and refractive errors, should be adjusted to decrease the risk of acquiring this condition. Online learning will remain, along with CVS, after this pandemic, and we hope our research will be taken into account in remodelling our education system accordingly.

## BACKGROUND

In 2019, there was a report from Wuhan Municipal Health Commission about a cluster of cases of pneumonia in Wuhan, China.[1–3] What started as an outbreak in China is now a global crisis. On 11 March 2020, the WHO announced that COVID-19 could be classified as a pandemic.[4] Suppression measures aimed

### What is known about the subject?

⇒ Computer vision syndrome (CVS) describes a group of vision-related problems caused by prolonged use of digital devices.
⇒ Learning has shifted from being performed in schools to taking place online during the COVID-19 pandemic.

### What this study adds?

⇒ Digital device use has increased by 4 hours per day, and the mean hours of online learning was 7 hours per day.
⇒ CVS was found in 70.1% of students.
⇒ Factors associated with CVS included digital device usage >6 hours/day, multiple digital device usage, refractive errors, and presence of back and neck pain.

at slowing down epidemic growth by reducing the number of cases and human-to-human transmission included social distancing and closure of schools and universities, and[5 6] UNESCO reported that 1.37 billion students from over 130 countries were affected by these interventions. For Thailand, the total duration of school closure was 42 weeks.[7]

Many schools shifted from classroom-based learning to online schooling in order to continue teaching. Up to 54% of parents of children aged 5–15 years reported as many as 5 additional hours spent online.[8] Moderate use of screens (4 hours/day) was associated with lower psychological well-being, including less curiosity, lower self-control, distractibility, and inability to finish tasks.[9]

Rapid advancement in technology has led to digital devices becoming a big part of our daily lives, especially for students, who use digital devices as homework aids, for

reading, and for leisure activities. Approximately 40% of school pupils have been found to spend less than 2 hours a day or 2–4 hours a day on digital device reading, while 14% spent 4–6 hours and 3% spent over 6 hours each day using digital devices for reading.[10] Prolonged screen time can produce physical discomfort known as digital eye strain or computer vision syndrome (CVS). The American Optometric Association defined CVS as a group of eye and vision-related problems that result from prolonged usage of digital devices which cause increased stress to near vision.[11] The diagnosis of CVS is subjective, with numerous questionnaires being developed to diagnose this syndrome, including the 17-item Computer-Vision Symptom Scale questionnaire, a six-item visual fatigue scale by Benedetto *et al,* and the Computer Vision Syndrome Questionnaire (CVS-Q) by Segui *et al.*[12–14] The CVS-Q is a validated questionnaire commonly used in clinical trials to evaluate the visual health of digital device users.

Before the COVID-19 era, the prevalence of CVS among university students, adults, and office workers was between 60% and 80%.[15] Eye strain was found in 18% of teenage students at the end of the day after working on digital devices.[10] A study from Indonesia reported that 87.2% of high school students experienced evaporative dry eye, which is one of the risk factors of CVS.[16] On average, children aged 8–12 years in the USA were found to spend 4–6 hours a day watching or using screens, while teens spent up to 9 hours.[17] In Thailand, 94.84% of secondary school students were found to have at least one symptom of CVS in 2016. Thai children spend approximately 35 hours per week watching screens.[18]

The prevalence of dry eye symptoms is greater during electronic screen use than when viewing printed materials.[19] Environmental factors such as use of air conditioning and windy environments have been reported to correlate with visual symptoms of dry eye disease,[20] which is a major contributor to CVS.[21] Factors previously reported to be associated with CVS were hours of use, screen distance, screen brightness, room illumination, wearing of contact lenses, and refractive errors.[10 22] Commonly reported symptoms relating to CVS are headaches, eye strain, blurred vision, dry eye symptoms, and pain in the neck and shoulders.

This study aims to assess the prevalence and associated factors of CVS among school students in Bangkok during lockdown resulting from the COVID-19 pandemic.

## MATERIALS AND METHODS

An online questionnaire was sent to primary and secondary school students (ages 10–19 years) electronically. Before answering the questionnaire, all participants were informed on the cover letter of the questionnaire about the study's purpose, methods, and guarantee of anonymity of data. All participants were required to sign and accept an informed consent form before continuing with the survey, in which they answered the questions themselves. There were no patient and public involvement in this study. Data were collected between 16 August 2021 and 31 August 2021 (15 days), during online schooling in accordance with the COVID-19 lockdown policy.

## Online questionnaire

The questionnaire consisted of four parts: demographic data and electronic device usage before the pandemic; online learning behaviour; a CVS-Q; and a poster providing students with information on the proper use of electronic devices (online supplemental file 1).

The CVS-Q, which was developed by Segui *et al,*[14] investigates the presence of 16 eye symptoms. High school students were required to report the frequency and intensity of each eye symptom. For each symptom, the frequency score is multiplied by the intensity score and all of the scores for each symptom are added together. For the person to be considered as having CVS, he or she must have a total score of greater than or equal to 6. The severity was divided into mild, moderate and severe, corresponding with scores of 6–12, 13–18, and 19 or over, respectively.

## Statistical analysis

All the data from the electronic survey were analysed with SPSS V.16.0 for Windows. Frequencies and percentages were used for categorical data. Continuous data were reported using mean, median, and SD after confirmation of normal distribution of the data. Paired t-test was used to compare the number of hours of digital device use during and prior to the COVID-19 pandemic, while one-way analysis of variance was used to compare hours of online learning, total hours of digital device usage, and severity of CVS. Risk factors associated with CVS were analysed by univariate and multiple logistic regression to identify independent risk factors of CVS by calculating the OR and their corresponding 95% CI. All variables with a p value of <0.05 in the univariate analysis were further analysed by multiple logistic regression. A p value of <0.05 was considered statistically significant.

## RESULTS
### Baseline characteristics

A total of 2476 students completed the online survey. Demographic data and electronic device usage before the pandemic are shown in table 1. The participants' mean age was 15.52±1.66 years with a female predominance (64.9%). The majority of students were in grades 10–12 (68%), and around 40% used glasses or contact lenses to correct their refractive errors. Mobile phones were the most used digital device before the COVID-19 pandemic at 54.2%, and a quarter of the students had around 5–6 hours of screen time before the lockdown (table 2).

**Table 1** Demographic characteristics of participants and usage of electronic devices before the pandemic (N=2476)

| Demographic characteristics | N (%) |
|---|---|
| Mean age (years) ±SD | 15.52±1.66 (range 10–19) |
| Female | 1606 (64.9) |
| Grade | |
| 4–6 | 11 (0.4) |
| 7–9 | 781 (31.6) |
| 10–12 | 1684 (68) |
| Refractive status | |
| Emmetropia | 1180 (47.7) |
| Myopia | 1055 (42.6) |
| Hyperopia | 241 (9.7) |
| Glasses and contact lens (CL) use | |
| None | 1180 (47.7) |
| Glasses for myopia | 784 (31.7) |
| Glasses for hyperopia | 186 (7.5) |
| CL for myopia | 151 (6.1) |
| CL for hyperopia | 47 (1.9) |
| Glasses and CL for myopia | 120 (4.8) |
| Glasses and CL for hyperopia | 8 (0.3) |
| Most commonly used digital devices (pre-COVID-19) | |
| Mobile phone | 1341 (54.2) |
| Tablet | 454 (18.3) |
| Computer desktop | 436 (17.6) |
| Computer laptop | 216 (8.7) |
| Television | 29 (1.2) |
| Frequency of eye check-up | |
| None | 1054 (42.6) |
| Once every 2 years | 504 (20.4) |
| Once a year | 715 (28.9) |
| Twice a year | 203 (8.2) |

### Digital device usage during the COVID-19 pandemic

The mean numbers of hours spent per day using digital devices before and after the pandemic were 6.13±2.8 and 10.53±2.99, respectively (p<0.001) (table 2). Over 60% of the students spend at least 7 hours/day with online learning (mean=7.03±2.06 hours/day), and 40% of them used multiple devices when learning online, with mobile phones being the most used. The majority of students did their online studying in a fan-ventilated environment, and over 80% employed some sort of protective equipment, including blue-coated glasses or protective films on their digital devices. Symptoms of back pain and neck pain were present in around 75.9% and 68.1% of students respectively during lockdown (table 3).

**Table 2** Comparison between duration of digital device usage per day before and during COVID-19 lockdown (N=2476)

| Duration of digital device usage per day | Before COVID-19 | During COVID-19 |
|---|---|---|
| ≤2 hours | 201 (8.1%) | 67 (2.7%) |
| 3–4 hours | 596 (24.1%) | 271 (21.9%) |
| 5–6 hours | 648 (26.2%) | 502 (20.3%) |
| 7–8 hours | 492 (19.9%) | 1109 (44.8%) |
| 9–10 hours | 401 (16.2%) | 480 (19.4%) |
| 11–12 hours | 49 (1.9%) | 16 (0.7%) |
| >12 hours | 89 (3.6%) | 31 (1.3%) |
| **Number of hours spent per day (mean±SD)** | 6.13±2.8 | 10.53±2.99 |

### Computer Vision Syndrome Questionnaire

The median CVS-Q score was 11 (range 0–64), and 70.1% of participants had CVS. Increasing severity of the condition was correlated with the number of hours of online learning and total hours of digital device usage (p<0.001) (figure 1).

The most common symptoms from CVS-Q were headaches (n=1921, 77.58%), burning (n=1791, 72.33%) and eye pain (n=1767, 71.37%). In terms of severity, headaches were the most severe symptom (n=392, 15.83%), followed by worsening of eyesight (n=266, 10.74%) and pain (n=256, 10.34%) (figure 2).

### Associated factors

Univariate analysis revealed multiple factors associated with CVS, including female gender (p≤0.001) and device usage without additional protection (blue-coated glasses, blue-coated film, or both) (p=0.006). Students who did their online studying with fans had a higher risk of developing CVS in comparison with students who studied in an air-conditioned environment (p<0.001), and those who preferred using laptop computers had the greatest risk of developing CVS (OR=2.36, 95% CI: 1.75 to 3.19, p<0.001) (table 4).

Multiple logistic regression analysis showed that the significant factors associated with CVS were age ≤15 years (adjusted OR (AOR)=2.17, 95% CI: 1.36 to 3.45, p=0.01), digital device usage >6 hours/day (AOR=1.91, 95% CI: 1.13 to 3.23, p=0.016), online learning >5 hours/day (AOR=4.99, 95% CI: 3.08 to 8.12, p<0.001), multiple digital device usage for online learning (AOR=2.15, 95% CI: 1.04 to 4.43, p=0.038), refractive errors (AOR=2.89, 95% CI: 1.83 to 4.54, p<0.001), presence of back pain (AOR=2.06, 95% CI: 1.32 to 3.22, p=0.001), and presence of neck pain (AOR=2.36, 95% CI: 1.89 to 3.70, p<0.001). Myopia and emmetropia were independent risk factors (AOR=2.11, 95% CI: 1.24 to 3.32 and AOR=2.09, 95% CI: 2.14 to 3.47, respectively, p<0.001).

 

**Table 3** The usage of electronic devices in online learning during COVID-19 lockdown (N=2476)

| Demographic characteristics | N (%) |
|---|---|
| Total hours of online learning | |
| ≤2 hours | 67 (2.7) |
| 3–4 hours | 271 (11) |
| 5–6 hours | 502 (20.3) |
| 7–8 hours | 1109 (44.8) |
| 9–10 hours | 480 (19.4) |
| >10 hours | 47 (1.8) |
| Device used for online learning | |
| Single device | 1486 (60) |
| Multiple device | 990 (40) |
| Mobile phone | 1174 (47.4) |
| Tablet | 1095 (44.2) |
| Computer desktop | 965 (39) |
| Computer laptop | 691 (27.9) |
| Television | 27 (1.1) |
| Environment | |
| Fan | 1272 (51.4) |
| Air conditioning | 1203 (48.6) |
| Protective instrument | |
| None | 413 (16.7) |
| Blue-coated glasses | 1172 (47.3) |
| Blue-coated film on digital devices | 680 (27.5) |
| Both blue-coated glasses and film on digital device | 211 (8.5) |
| Frequency of eye rest | |
| Never | 371 (15) |
| Every 15 min | 596 (24.1) |
| Every 30 min | 431 (17.4) |
| Every 45 min | 304 (12.3) |
| Every 1 hour | 427 (17.2) |
| Every ≥1 hour | 347 (14) |
| Distance of digital devices from eyes during online learning | |
| <40 cm | 943 (38.1) |
| 40–80 cm | 1260 (50.9) |
| >80 cm | 273 (11) |
| Back pain | 1880 (75.9) |
| Neck pain | 1687 (68.1) |
| Activity when resting | |
| Close eyes | 1144 (46.2) |
| Sleep | 1230 (49.7) |
| Look out | 822 (33.2) |
| Play games | 568 (22.9) |
| Use artificial tears | 271 (10.9) |

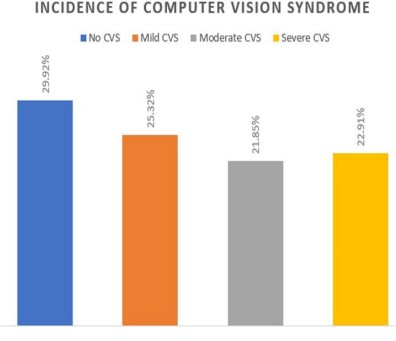

INCIDENCE OF COMPUTER VISION SYNDROME

■ No CVS  ■ Mild CVS  ■ Moderate CVS  ■ Severe CVS

29.92%  25.32%  21.85%  22.91%

| | No CVS (n=741) | Mild CVS (n=627) | Moderate CVS (n=541) | Severe CVS (n=567) | P-Value |
|---|---|---|---|---|---|
| Mean number of hours per day of online learning (hours) ± SD | 6.16 ± 2.26 | 7.23 ± 2.01 | 7.28 ± 1.78 | 7.69 ± 1.74 | < 0.001 |
| Mean hours per day of total digital device usage (hours) ± SD | 9.45 ± 3.38 | 10.77 ± 2.93 | 10.89 ± 2.57 | 11.33 ± 2.54 | < 0.001 |

**Figure 1** The mean number of hours per day of online learning and total hours per day of digital device usage according to computer vision syndrome (CVS) severity.

classroom environment. Digital device usage during this period increased by over 4 hours per day, with a mean of 7 hours per day of online learning. The prevalence of CVS in Thai students was 70.1%, and its severity correlated with the number of hours of digital device usage and the number hours of online learning. The most common symptoms of CVS were headaches (77.58%), burning (72.33%), and eye pain (71.37%). Multiple digital device usage during online learning, refractive error, and the presence of neck and back pain, were also independent associated factors of CVS.

Increased online learning has been reported in all levels of education.[23 24] Students previously used digital devices as homework aids and for reading textbooks, but after school closures, they were used for online classes as well, adding to the number of hours of screen time per day. The increase in hours spent on digital devices was mostly for educational purposes.[25] During virtual learning due to the COVID-19 pandemic, 50% of students in India had

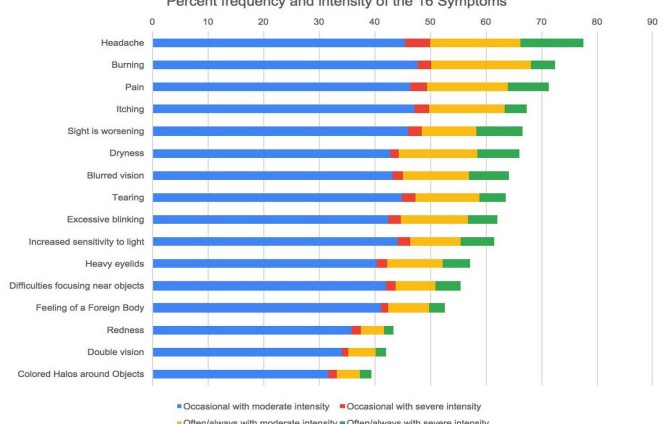

**Figure 2** The percentage of frequency and severity of different eye symptoms of computer vision syndrome in students.

## DISCUSSION

During the COVID-19 pandemic, a lockdown strategy was used to control the spread of the disease, so schools were closed and online learning replaced the normal

**Table 4** Multiple logistic regression showing the associated risk factors of CVS

| | Univariate analysis | | | Multiple analysis | | |
|---|---|---|---|---|---|---|
| **Factor** | **Crude OR** | **95% CI** | **P value** | **Adjusted OR** | **95% CI** | **P value** |
| Age ≤15 | 2.34 | 1.63 to 3.36 | <0.001 | **2.17** | **1.36 to 3.45** | **0.01** |
| Female | 1.73 | 1.34 to 2.23 | <0.001 | 1.26 | 0.90 to 1.75 | 0.178 |
| Overall digital usage >6 hours | 7.41 | 5.52 to 9.96 | <0.001 | **1.91** | **1.13 to 3.23** | **0.016** |
| Online learning >5 hours | 7.99 | 6.07 to 10.53 | <0.001 | **4.99** | **3.08 to 8.12** | **<0.001** |
| Refractive error | 1.35 | 1 to 02 to 1.79 | 0.035 | **2.89** | **1.83 to 4.54** | **<0.001** |
| Refractive error | | | | | | |
| Myopia | 3.21 | 2.31 to 4.44 | <0.001 | **2.11** | **1.24 to 3.32** | **<0.001** |
| Emmetropia | 3.19 | 2.21 to 4.60 | <0.001 | **2.09** | **2.14 to 3.47** | **<0.001** |
| Hyperopia | Ref | | | **Ref** | | |
| Fan | 2.55 | 1.86 to 3.48 | <0.001 | 1.2 | 0.81 to 1.81 | 0.362 |
| Non-protective device used | 1.82 | 1.19 to 2.79 | 0.006 | 1.07 | 0.66 to 1.73 | 0.793 |
| Protective device | | | | | | |
| Both | 4.62 | 2.90 to 7.38 | <0.001 | 0.66 | 0.39 to 1.11 | 0.118 |
| Blue-coated glasses | 3.36 | 2.52 to 4.49 | <0.001 | 1.19 | 0.71 to 1.99 | 0.508 |
| Blue-coated film | Ref | | | Ref | | |
| Rest of over 45 min | 2.59 | 2.01 to 3.34 | <0.001 | 1.02 | 0.7 to 1.48 | 0.935 |
| Other digital devices used >2 hours | 2.22 | 1.73 to 2.85 | <0.001 | 0.89 | 0.61 to 1.28 | 0.52 |
| Distance from device <40 cm | 2.42 | 1.78 to 3.30 | <0.001 | 1.07 | 0.71 to 1.63 | 0.743 |
| Back pain | 2.84 | 2.12 to 3.80 | <0.001 | **2.06** | **1.32 to 3.22** | **0.001** |
| Neck pain | 2.59 | 1.94 to 3.46 | <0.001 | **2.64** | **1.89 to 3.70** | **<0.001** |
| Multiple digital devices used | 3.6 | 2.76 to 4.70 | <0.001 | **2.15** | **1.04 to 4.43** | **0.038** |
| Non-use of artificial tears | 0.96 | 0.70 to 1.30 | 0.77 | 1.34 | 0.91 to 1.98 | 0.133 |
| Close eyes during online learning | 1.69 | 1.32 to 2.17 | <0.001 | 1.12 | 0.80 to 1.55 | 0.508 |
| Sleep during online learning | 2.12 | 1.64 to 2.74 | <0.001 | 1.29 | 0.93 to 1.79 | 0.135 |
| Laptop computer preference | 2.36 | 1.75 to 3.19 | <0.001 | 0.83 | 0.48 to 1.42 | 0.494 |
| Tablet preference | 2.27 | 1.77 to 2.92 | <0.001 | 0.86 | 0.51 to 1.45 | 0.568 |
| Television screen preference | 1.49 | 0.49 to 4.55 | 0.49 | 1.067 | 0.30 to 3.75 | 0.92 |
| Computer desktop preference | 1.37 | 1.07 to 1.77 | 0.016 | 1 | 0.61 to 1.66 | 0.987 |
| Mobile phone preference | 1.34 | 1.03 to 1.69 | 0.031 | 0.81 | 0.50 to 1.31 | 0.395 |

CVS, computer vision syndrome.

CVS, while 77% of students in China reported having at least one of its symptoms.[25–27]

Among those with CVS, the severity was significantly correlated with the number of hours of screen time, with the use of electronic devices for over 5 hours found to be associated with it.[22] The use of mobile touch screen devices was related to the development of musculoskeletal symptoms.[28] The most common symptom of CVS in our study was headache, in line with the findings of many other published reports.[22 27 29]

Students under 15 years were twice as likely to have CVS as their older counterparts, with the higher severity occurring in younger students. Prevalence of CVS was found to be lower in older Spanish university students than in the younger group.[29] In contrast, a Chinese study found that CVS was independently associated with older age.[26] These apparent anomalies may be attributed to variations in the mean age of students who have diverse levels of digital devices used. The number of hours of digital device usage varied across grades.[30] Another possible explanation for diverse correlations with age may be the developmental differences in age groups, as younger students might have had problems answering some parts of the questionnaire. A previous study found that children and adolescents aged 6–17 years were unable to report symptoms of dry eye correctly.[31]

Multiple digital device usage was another independent associated factor of CVS. Each type of device has its own recommended viewing distance. For computers, the reading and writing distance is usually 30–40 cm from the eyes, and less eye strain was found when the computer monitor was 50–70 cm from the eyes.[32] Mobile phones

and tablets, which have smaller screens, are usually held closer, at about 20–30 cm from the eyes.[33] Half of the students in this study reported their reading distance as between 40 and 80 cm, which is the appropriate distance for computers. Only 30% and 38.9% of students, respectively, used laptop and desktop computers, while about half used multiple devices. When alternating between devices, students may not adjust the viewing distance appropriately, and this could lead to symptoms of CVS. Laptop computers, tablets and smartphones are typically held in downward gaze, and greater corneal exposure from higher gaze angles results in increased tear evaporation. Variations in gaze position when alternating between devices can lead to CVS.[19]

Just as different types of digital devices require different viewing positions, they also involve variations in posturing, and improper ergonomics can lead to neck and back pain, which were independent risk factors in our study, in agreement with the findings of earlier reports.[22 34 35] A previous study found that a pattern of using smartphones or tablets in bouts of 1 hour or more carried a higher risk of musculoskeletal symptoms than the total duration of use throughout the day.[28] Some students have reported lying on the bed while studying.[10]

Having refractive error, particularly myopia, adds to the risk of developing CVS. Similar results were found in research in China which concluded that students with self-reported myopia, both who did and did not wear glasses, were at higher risk of CVS compared with those who were not myopic.[26] Children with myopia could possibly have residual near work-induced transient myopia from impaired sympathetic function, eventually leading to permanent myopic progression.[36] Long hours of online studying might lead to myopic progression, causing children to wear undercorrected lenses, and residual refractive errors combined with continued near-work studying have been found to lead to CVS.

Students with myopia and hyperopia in our study were those who reported wearing myopic or hyperopic corrective lenses for their refractive errors; those classified as emmetropic (28.8%) reported not using corrective lenses because our questionnaire asked whether students wore glasses or contact lenses for short-sightedness or long-sightedness, not whether they had refractive errors. Around 40% of students reportedly never had eye examinations, while 20% receive one every 2 years. Students with some uncorrected refractive errors could be hidden in this group, which may be the reason why emmetropia was a significant risk factor for developing CVS. People with uncorrected refractive errors have been found to be at a higher risk than those with corrected refractive errors and those without refractive errors.[26]

Prolonged exposure to computers in patients with CVS has been reported to significantly correlate with dry eye disease.[37] Environmental factors producing corneal drying include low ambient humidity, high forced-air heating or air conditioning settings, together with the use of ventilation fans, excess static electricity,

or airborne contaminants.[38] A study among information technology professionals in Egypt found that exposure to air pollution, use of air conditioners, and exposure to windy environments, were significant predictors of CVS.[39] Symptoms of dry eye are major components of CVS[40]; therefore, identifying environmental factors related to dry eye is necessary to make adjustments to minimise the condition.

This study provides comprehensive data on CVS during the COVID-19 pandemic from high school students using a validated questionnaire. Not only students, but also parents, teachers, and schools could benefit from using this study in terms of developing appropriate guidelines for online learning. Timetables should be adjusted to have appropriate duration and breaks. The overall digital device usage should be under 6 hours/day, and online learning should be limited to 5 hours/day, especially in younger students. This study emphasises the need for regular eye examinations for students, especially those with refractive errors, which should be fully corrected. Proper ergonomics and learning environments are also important; students should adjust posturing and viewing distance according to the digital device used.

The main limitation of our study is that it used a self-reported questionnaire which is subject to bias. Due to the cross-sectional nature of this research, we were only able to identify factors associated with CVS, which has been defined by a mixture of symptoms and signs resulting in various definitions being used across clinical research, thereby limiting the available reports that could be compared with our results. The participants in this study were from a single city, so that it does not represent the online schooling situation in all of Thailand. Lastly, we did not have students perform the CVS-Q before the pandemic, so we do not have data for comparison of CVS before and during lockdown.

Further evaluations with objective methods such as tear break-up time, Schirmer test, and ocular surface staining could be added into the analysis. A possible pattern of online learning and its association with CVS could be investigated in a study of longer duration.

## CONCLUSION

During the pandemic, students have spent an increasing number of hours on digital devices, and over 70% have CVS, with headaches being the most frequent symptom. The number of hours spent on digital devices and online learning, refractive errors, multiple digital device usage, presence of back pain, presence of neck pain, and younger age are factors associated with CVS.

Online learning had already grown over the years before the COVID-19 pandemic, but it has expanded exponentially during times of social distancing. We believe that even after this pandemic, online learning will remain, along with CVS. Our study points out factors associated with this condition which we hope will be

taken into consideration in remodelling our education system appropriately.

## Author affiliations

[1]Sirindhorn International Institute of Technology, Thammasat University, Khlong Nueng, Pathum Thani, Thailand
[2]Department of Ophthalmology, College of Medicine, Rangsit University, Rajavithi Hospital, Bangkok, Thailand
[3]Department of Ophthalmology, Walailak University Hospital, Walailak University, Nakhon Si Thammarat, Thailand
[4]Department of Tropical Hygiene, Faculty of Tropical Medicine, Mahidol University, Bangkok, Thailand
[5]Department of Ophthalmology, Mettapracharak(Wat Rai Khing) Hospital, Nakhon Pathom, Thailand
[6]Faculty of Medicine, Chulalongkorn University, Bangkok, Thailand
[7]Royal Society of Thailand, Bangkok, Thailand

**Contributors** KS conceived and designed the study, collected, analysed and interpreted the data, wrote the first draft of the manuscript and is responsible for the overall content as guarantor. Conceptualisation—KS and TT. Data collection—KS, WS and NY. Methodology—KS, WT, NS, PS and TT. Statistical analysis—KS, NS and TT. Writing (original draft)—KS, WT, NS, PS, NY and TT. Writing (review and editing)—KS, WT, WS, NS, PS, NY and TT.

**Funding** The authors have not declared a specific grant for this research from any funding agency in the public, commercial or not-for-profit sectors.

**Competing interests** None declared.

**Patient and public involvement** Patients and/or the public were involved in the design, or conduct, or reporting, or dissemination plans of this research. Refer to the Methods section for further details.

**Patient consent for publication** Not required.

**Ethics approval** This cross-sectional, online, questionnaire-based study was approved by the institutional review boards of Rajavithi Hospital, Thailand (ID 64237), and it was conducted in accordance with the tenets of the Declaration of Helsinki. Participants gave informed consent to participate in the study before taking part.

**Provenance and peer review** Not commissioned; internally peer reviewed.

**Data availability statement** Data are available upon reasonable request.

**ORCID iD**
Thanaruk Theeramunkong http://orcid.org/0000-0003-0744-4213

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
