## [Reviewer comments · BMJ Paediatrics Open]

ARTICLE DETAILS

TITLE (PROVISIONAL)	Effects of digital devices and online learning on computer vision syndrome in students during the COVID-19 era
AUTHORS	Seresirikachorn, Kasem Thiamthat, Warakorn Sriyuttagrai, Wararee Soonthornworasiri, Ngamphol Singhanetr, Panisa Yudtanahiran, Narata Theeramunkong, Thanaruk

VERSION 1 – REVIEW

REVIEWER	Reviewer name: Dr. Eirini Koutoumanou Institution and Country: University College London, United Kingdom of Great Britain and Northern Ireland Competing interests: None
REVIEW RETURNED	09-Mar-2022

GENERAL COMMENTS	- I recommend that the manuscript is reviewed once more for grammatical errors, such as "should be adjusted to decreased", "it's downside" – there are possibly more. Also, the title of Figure 1 needs editing as it does not read correctly- Some of the variables of interest have been turned into categorical format instead of their original numerical version. Did the authors attempt to analysis those variables in numerical format?- Were the 15 days of the data collection duration during school term? It is mentioned that data were collected during school closure, but I think this point needs to be clarified as in the UK, for example, the term "school closure" is often used for holiday term, unlike school closure due to covid19 during term time?- Are grades 4 to 12 in Thailand equivalent to years 4 and 12 in the UK? Could this point be clarified please and for comparison purposes across counties, refer to the age of the students in the text too?- Could the authors please clarify if it was the students or the parents that completed the questionnaire?- "Descriptive statistics were used for categorical data" – descriptive statistics include means, sd etc for numerical data too. So, this should be rephrased to say "Frequencies and percentages were used for categoric data..."- Was the Normality of numeric variables checked? Means and medians reported but in terms of spread just SD, and not IQR. Why is that?- It is said that multivariate logistic regression was used but there are no multiple outcomes, which is what multivariate stands for (see reference below). Instead, when multiple predictors are included in a model, the model is multiple or multivariable. This needs to be
---

	corrected in all relevant parts in the paper included Tables.  - Were all the variables in the multiple logistic model shown in table 3 added in the model at once? So the adjusted OR are adjusted for all variables listed in this table? Could you please clarify in the text? - The statistically significant differences shown under Figure 1 are actually comparing means that are only slightly different between each other. Can the authors comment on the magnitude of these differences? Are these deemed clinically important? E.g. mean age ranging from 15.10 to 15.55. - The authors should acknowledge few more limitations regarding their study design and data collection process too. These include the risk of potential biases because of the retrospective, online and self-reported nature of the collected data. Also, was 2 weeks of data enough time to capture a potential pattern/association? - The authors need to rephrase any concluding statements like the following that imply that some factor contribute to developing CVS: "...are contributory factors of developing CVS". The observational nature of this study is only able to identify associations between CVS and symptoms but not to identify what might be leading/contributing to developing CVS. References: Hidalgo, Bertha, and Melody Goodman. "Multivariate or multivariable regression?." American journal of public health 103.1 (2013): 39-40.
--	--

VERSION 1 – AUTHOR RESPONSE

Dear Editor-in-Chief,

We would like to thank the Editor for your helpful comments. Please kindly find the response of each suggestion or comment as follow.

Editor in Chief Comments to Author :

Title add "an online questionnaire study"

Use "mean" instead of "average" throughout the paper.

Create a table comparing "Duration of digital device usage" and "Type of digital device used" before and during COVID. At present the data are shown separately in Tables 1 and 2 making comparison difficult.

Suggestion, Question, or Comment from the Editor	Author's Response	Change in the Manuscript
1. Title add "an online questionnaire study"	We have change the tittle to "Effects of digital devices and online learning on computer vision syndrome in students during the COVID-19 era:	Page 1 line 2-3, 25 Page 2 line 2

	an online questionnaire study”	
2. Use "mean" instead of "average" throughout the paper.	We have replaced “mean” instead of “average” throughout the paper	Page 2 line 11 Page 3 line 8 Page 6 line 17 Page 7 line 25
3. Create a table comparing "Duration of digital device usage" and "Type of digital device used" before and during COVID. At present the data are shown separately in Tables 1 and 2 making comparison difficult.	We have created a table comparing the duration of digital device usage before and during COVID and labeled it Table 2. Types of digital device used could not be compared because before COVID we asked what was the most frequently used digital device in general, but during COVID we asked what was the most frequently used device for online learning.	Table 2

Associate Editor

Comments to the Author:

I think this is an important study.

it comments on an obvious finding of the shift to technological distance learning, but the study is clearly limited by methods. I think the authors need to be clearer on this in three major ways:

- existing prevalence data and diagnostic consensus. Details of any consensus criteria for diagnosis, the gold standard for doing this or comments on the lack of this are key. Also, what have previous studies in school children said? When was it first reported? What essentially is the pre-pandemic baseline. Also, what does literature say about pre-pandemic screen use. Some is in the discussion, but more specifics are needed and this should be in the background, not the discussion.

- Consideration of literature on risk factors. Key risks that were interesting such as fans vs

air conditioning are key, but aren't contextualised in the background or discussion and this is key. It seems the results are not written concordantly with these sections.

- Limitations - these are far more and need referenced details on the limitations of such a cross sectional methodology. Far more details on the next step for future research and practice would link to this

With these minor - and ultimately not methods - changes, this work will be a delight to review again.

Suggestion, Question, or Comment from the Editor	Author's Response	Change in the Manuscript
1. existing prevalence data and diagnostic consensus. Details of any consensus criteria for diagnosis, the gold standard for doing this or comments on the lack of this are key. Also, what have previous studies in school children said? When was it first reported? What essentially is the pre-pandemic baseline. Also, what does literature say about pre-pandemic screen use. Some is in the discussion, but more specifics are needed and this should be in the background, not the discussion.	CVS diagnosis has been clarified in the text. We have added the reported pre-pandemic baseline in Thailand and other countries and we have moved this information into the background.	Page 4 line 3-10 Page 4 line 11-18
2. Consideration of literature on risk factors. Key risks that were interesting such as fans vs air conditioning are key, but aren't contextualised in the background or discussion and this is key. It seems the results are not written concordantly with these sections.	We have added previously reported papers that mentioned fans and air conditioning environment in CVS in the background and discussed it in the text as well.	Page 4 lines 19-26 Page 10 line 1-8
3. Limitations - these are far more and need referenced details on the limitations of	We add some limitations about the nature of this cross sectional study	Page 10 lines 18-28

such a cross sectional methodology. Far more details on the next step for future research and practice would link to this	and future directions of our study.	
---	-------------------------------------	--

Reviewer: 1

Dr. Eirini Koutoumanou, University College London

Comments to the Author

- I recommend that the manuscript is reviewed once more for grammatical errors, such as “should be adjusted to decreased”, “it’s downside” – there are possibly more. Also, the title of Figure 1 needs editing as it does not read correctly
- Some of the variables of interest have been turned into categoric format instead of their original numerical version. Did the authors attempt to analysis those variables in numerical format?
- Were the 15 days of the data collection duration during school term? It is mentioned that data were collected during school closure, but I think this point needs to be clarified as in the UK, for example, the term “school closure” is often used for holiday term, unlike school closure due to covid19 during term time?
- Are grades 4 to 12 in Thailand equivalent to years 4 and 12 in the UK? Could this point be clarified please and for comparison purposes across counties, refer to the age of the students in the text too?
- Could the authors please clarify if it was the students or the parents that completed the questionnaire?
- “Descriptive statistics were used for categorical data” – descriptive statistics include means, sd etc for numerical data too. So, this should be rephrased to say “Frequencies and percentages were used for categoric data...”
- Was the Normality of numeric variables checked? Means and medians reported but in terms of spread just SD, and not IQR. Why is that?
- It is said that multivariate logistic regression was used but there are no multiple outcomes, which is what multivariate stands for (see reference below). Instead, when multiple predictors are included in a model, the model is multiple or multivariable. This needs to be corrected in all relevant parts in the paper included Tables.
- Were all the variables in the multiple logistic model shown in table 3 added in the model at once? So the adjusted OR are adjusted for all variables listed in this table? Could you please clarify in the text?
- The statistically significant differences shown under Figure 1 are actually comparing

means that are only slightly different between each other. Can the authors comment on the magnitude of these differences? Are these deemed clinically important? E.g. mean age ranging from 15.10 to 15.55.

- The authors should acknowledge few more limitations regarding their study design and data collection process too. These include the risk of potential biases because of the retrospective, online and self-reported nature of the collected data. Also, was 2 weeks of data enough time to capture a potential pattern/association?

- The authors need to rephrase any concluding statements like the following that imply that some factor contribute to developing CVS: "...are contributory factors of developing CVS". The observational nature of this study is only able to identify associations between CVS and symptoms but not to identify what might be leading/contributing to developing CVS.

Suggestion, Question, or Comment from the Editor	Author's Response	Change in the Manuscript
1. I recommend that the manuscript is reviewed once more for grammatical errors, such as "should be adjusted to decreased", "it's downside" – there are possibly more. Also, the title of Figure 1 needs editing as it does not read correctly	We used an English corrector to help us review and correct the grammar for this manuscript before re-submitting it.	All manuscript
2. Some of the variables of interest have been turned into categoric format instead of their original numerical version. Did the authors attempt to analysis those variables in numerical format?	We have analyzed the data in numerical format, but we chose the categorical format since we think that it has more clinical value. We wanted to find the duration of digital device use that was most correlated with CVS so that we could find a cut-off point to recommend screen time. Other studies found similar results in the recommended number of hours of digital devices used. -USA, Digital device usage should not be over 6 hours in children under 15 years ¹ -Report from The National Board of Professional Teaching advised less	

	than 5 hours per day for online learning ²	
3. Were the 15 days of the data collection duration during school term? It is mentioned that data were collected during school closure, but I think this point needs to be clarified as in the UK, for example, the term “school closure” is often used for holiday term, unlike school closure due to covid19 during term time?	We clarified the timeframe data were collected, which was during online schooling. Data was collected between August 16, 2021 and August 31, 2021 (15 days), during online schooling in accordance with the COVID-19 lockdown policy.	Page 5 lines 9-11
4. Are grades 4 to 12 in Thailand equivalent to years 4 and 12 in the UK? Could this point be clarified please and for comparison purposes across counties, refer to the age of the students in the text too?	We have changed the academic level to a more universal naming of “primary and secondary school”, and added the age range into the text.	Page 5 line 5
5. Could the authors please clarify if it was the students or the parents that completed the questionnaire?	We clarified in the text that students completed the questionnaire themselves.	Page 5 lines 8-9
6. “Descriptive statistics were used for categorical data” – descriptive statistics include means, sd etc for numerical data too. So, this should be rephrased to say “Frequencies and percentages were used for categoric data...”	I rephrased to say “Frequencies and percentages were used for categorical data”	Page 5 lines 26

7. Was the Normality of numeric variables checked? Means and medians reported but in terms of spread just SD, and not IQR. Why is that?	We checked the normality of data before reporting the mean. Continuous data was reported using mean, median, and standard deviation (SD) after confirmation of normal distribution of the data.	Page 5 line 27-28
8. It is said that multivariate logistic regression was used but there are no multiple outcomes, which is what multivariate stands for (see reference below). Instead, when multiple predictors are included in a model, the model is multiple or multivariable. This needs to be corrected in all relevant parts in the paper included Tables.	We changed the text to multiple logistic regression	Page 2 line 15-16 Page 6 lines 2-3 Page 7 line 12 Table 4 line 1
9. Were all the variables in the multiple logistic model shown in table 3 added in the model at once? So the adjusted OR are adjusted for all variables listed in this table? Could you please clarify in the text?	We add text to clarify “All variables with a p-value < 0.05 in the univariate were further analyzed by multiple logistic regression.”	Page 6 lines 4-5
10. The statistically significant differences shown under Figure 1 are actually comparing means that are only slightly different between each other. Can the authors comment on the magnitude of these differences? Are these deemed clinically important? E.g. mean age ranging from 15.10 to 15.55.	After we analyzed with Post Hoc Tests it seem that significance was found only between “no CVS” and “severe CVS” so we decided to remove this data from Figure 1 since it is clinically insignificant. Aged below 15 was associated factor for CVS by multiple logistic analysis which shown in Table 4	Figure 1 Table 4

11. The authors should acknowledge few more limitations regarding their study design and data collection process too. These include the risk of potential biases because of the retrospective, online and self-reported nature of the collected data. Also, was 2 weeks of data enough time to capture a potential pattern/association?	We add the limitation about study design, study duration and potential biases.	Page 10 lines 18-28
12. The authors need to rephrase any concluding statements like the following that imply that some factor contribute to developing CVS: “...are contributory factors of developing CVS”. The observational nature of this study is only able to identify associations between CVS and symptoms but not to identify what might be leading/contributing to developing CVS.	We changed the text in the conclusion to clarify all factors are associated with CVS.	Page 2 lines 16,22 Page 3 line 11 Page 7 line 5 Page 8 line 25 Page 11 lines 6, 9-10

Reference

1. American Academy of Child & Adolescent Psychiatry. Screen Time and Children. Available at: [https://www.aacap.org/AACAP/Families and Youth/Facts for Families/FFF-Guide/Children-And-Watching-TV054.aspx#:text=On%20average%2C%20children%20ages%208,use%20may%20lead%20to%20problems](https://www.aacap.org/AACAP/Families_and_Youth/Facts_for_Families/FFF-Guide/Children-And-Watching-TV054.aspx#:text=On%20average%2C%20children%20ages%208,use%20may%20lead%20to%20problems). Accessed April 8, 2022.
2. The National Board of Professional Teaching Standard. What are industry standards for time Spent Learning online. Available at: <https://eduww.net/how-much-time-should-students-spend-studying-online/>. Accessed April 8, 2022.

VERSION 2 – REVIEW

REVIEWER	Reviewer name: Institution and Country: Competing interests:
REVIEW RETURNED	

GENERAL COMMENTS	
--

REVIEWER	Reviewer name: Institution and Country: Competing interests:
REVIEW RETURNED	

GENERAL COMMENTS	
--

REVIEWER	Reviewer name: Institution and Country: Competing interests:
REVIEW RETURNED	

GENERAL COMMENTS	
--

VERSION 2 – AUTHOR RESPONSE

VERSION 3 – REVIEW

REVIEWER	Reviewer name: Institution and Country: Competing interests:
REVIEW RETURNED	

GENERAL COMMENTS	
--

REVIEWER	Reviewer name: Institution and Country: Competing interests:
REVIEW RETURNED	

GENERAL COMMENTS	
--

REVIEWER	Reviewer name: Institution and Country: Competing interests:
REVIEW RETURNED	

GENERAL COMMENTS	
--

VERSION 3 – AUTHOR RESPONSE